# The Feasibility of Stereotactic Body Proton Beam Therapy for Pancreatic Cancer

**DOI:** 10.3390/cancers14194556

**Published:** 2022-09-20

**Authors:** Hyunju Shin, Jeong Il Yu, Hee Chul Park, Gyu Sang Yoo, Sungkoo Cho, Joon Oh Park, Kyu Taek Lee, Kwang Hyuck Lee, Jong Kyun Lee, Joo Kyung Park, Jin Seok Heo, In Woong Han, Sang Hyun Shin

**Affiliations:** 1Department of Radiation Oncology, Samsung Medical Center, Sungkyunkwan University School of Medicine, Seoul 06351, Korea; 2Divisions of Hematology-Oncology, Department of Medicine, Samsung Medical Center, Sungkyunkwan University School of Medicine, Seoul 06351, Korea; 3Divisions of Gastroenterology, Department of Medicine, Samsung Medical Center, Sungkyunkwan University School of Medicine, Seoul 06351, Korea; 4Department of Surgery, Samsung Medical Center, Sungkyunkwan University School of Medicine, Seoul 06351, Korea

**Keywords:** pancreatic cancer, proton beam therapy, stereotactic body radiotherapy, dose escalation

## Abstract

**Simple Summary:**

Despite advances in treatment, the treatment outcome of pancreatic cancer still remains poor. Local progression can be a significant cause of several morbidities in pancreatic cancer, and dose escalation is needed. Stereotactic body proton beam therapy (SBPT) can give higher dose while minimizing dose at organ at risk with Bragg peak. The purpose of the present study was to investigate the feasibility of SBPT in pancreatic cancer. SBPT, administered in five fractions of a total 50–60 GyRBE, was performed mostly after induction chemotherapy. Grade 3 or higher gastroduodenal toxicities occurred in 6.1% of cases. The 2-year overall survival and local control rates were 67.6% and 73.0%. SBPT showed favourable treatment outcomes and treatment-related toxicities. It could be a promising alternative to radical surgery.

**Abstract:**

Background/Purpose: This study aimed to evaluate the clinical outcomes of stereotactic body proton beam therapy (SBPT) for pancreatic cancer. Methods: This retrospective study included 49 patients who underwent SBPT for pancreatic cancer between 2017 and 2020. Survival outcomes, bowel-related toxicities, and failure patterns were analysed. SBPT was performed after induction chemotherapy in 44 (89.8%) patients. The dose-fractionation scheme included 60 gray (Gy) relative biological effectiveness (RBE) in five fractions (*n* = 42, 85.7%) and 50 GyRBE in five fractions (*n* = 7, 14.3%). The median follow-up was 16.3 months (range, 1.8–45.0 months). Results: During follow-up, the best responses were complete response, partial response, and stable disease in four (8.2%), 13 (26.5%), and 31 (63.3%) patients, respectively. The 2-year overall survival, progression-free survival, and local control (LC) rates were 67.6%, 38.0%, and 73.0%, respectively. Grade ≥ 3 gastroduodenal (GD) toxicity occurred in three (6.1%) patients. Among them, one patient underwent endoscopic haemostasis. The other two patients received surgical management. They were followed up without disease progression for >30 months after SBPT. Overall, there was no significant dosimetric difference between the grade ≥ 2 and lower toxicity groups. Conclusions: SBPT provides relatively high LC rates with acceptable toxicities in pancreatic cancer.

## 1. Introduction

Pancreatic cancer is the seventh leading cause of cancer-related deaths worldwide and is predicted to become the third leading cause by 2025 [1]. Its prognosis remains poor, with a 5-year survival rate of approximately 10%, despite advances in pancreatic cancer treatment [2,3].

Currently, several modalities are used to treat pancreatic cancer. Although surgical resection is the backbone of management as the only curative treatment, treatment failure is common, even in early disease. Therefore, neoadjuvant and/or adjuvant therapies are recommended. Furthermore, in patients with unresectable disease, systemic chemotherapy with or without radiotherapy (RT), is generally adopted [4].

However, the systemic recurrence rate is approximately 50%, and the 5-year overall survival (OS) rate is only 25%, even after surgical resection with (neo) adjuvant therapy [5]. In locally advanced pancreatic cancer (LAPC), the distant metastasis rate within 3 months of diagnosis, even during initial treatment, is 30–50%. These facts emphasise the need for intensive systemic therapy. However, local progression (LP) is an important issue that cannot be overlooked. The local recurrence rate after curative surgical resection with (neo)adjuvant therapy has been reported to be 20–60% [6]. Additionally, systemic chemotherapy alone has been reported to have a higher LP rate, and 30–40% patients with pancreatic cancer die from locally progressive disease (PD) without distant metastases. Furthermore, LP can be a significant cause of several morbidities including biliary and/or gastroduodenal (GD) obstruction, bleeding, and perforation [7]. These findings also highlight the importance of combining RT with systemic therapy, thereby maximising local control (LC) in the treatment of pancreatic cancer, especially in unresectable disease [8].

The pancreas is surrounded by radiosensitive organs such as the duodenum, stomach, and small bowel. This makes it difficult to deliver sufficient radiation doses to pancreatic tumours. With advances in RT techniques, it has become possible to lower the dose to the surrounding organs and administer a higher dose to the pancreas. However, a high degree of caution is required for high dose administration to the pancreas, while simultaneously ensuring lower doses to other regions, and the treatment results need to be improved [9,10,11].

Stereotactic body RT (SBRT), which delivers a high dose in a few fractions, can shorten the treatment course, increase quality of life, minimize the interruption of intensive systemic therapy, and deliver a highly conformal ablative dose to the target with a sharp dose fall-off. This results in minimal exposure of at-risk organs. Proton beam therapy (PBT) offers a better dosimetric advantage. Protons travel a finite distance into the tissue determined by their energy and release most of that energy at a well-defined depth, forming a Bragg peak. Beyond the Bragg peak, there is little or no additional deposited dose. Consequently, radiation exposure to surrounding normal organs can be minimised [12]. Several studies have demonstrated that the radiation exposure of normal organs including liver, duodenum, and small bowel can be reduced using proton beam than photon beam in the treatment planning [12,13,14,15,16,17,18]. Based on the advantages of PBT, the application of stereotactic body PBT (SBPT) with chemotherapy is expected to improve LC in the management of pancreatic cancer. However, to date, limited studied have evaluated the treatment outcomes of SBPT in pancreatic cancer.

Therefore, we aimed to analyse the efficacy and safety of SBPT in pancreatic cancer. The relationship between GD toxicities and dose—volume histograms (DVHs) was analysed.

## 2. Materials and Methods

### 2.1. Patients

We retrospectively reviewed the medical records of patients with histologically confirmed pancreatic adenocarcinoma who underwent SBPT at our institution between January 2017 and December 2020. The institutional review board of Samsung Medical Center approved this retrospective study (no. 2021-07-038-001). We defined SBPT as the equivalent total dose delivered in 2 gray (Gy) fractions (EQD_2_) (alpha/beta ratio = 10 Gy) of ≥80 Gy relative biological effectiveness (RBE) in five fractions. Patients with a history of previous abdominal RT and those who received an incomplete planned SBPT course were excluded. Among the 65 patients with pancreatic cancer who underwent PBT during the study period, 49 met the inclusion criteria (Appendix A). The survival curves calculated from the start date of PBT to the last follow-up or event for all 65 patients are shown in Appendix A.

Each patient underwent an initial workup including physical examination, complete blood count analysis, standard blood chemistry evaluation, carbohydrate antigen (CA) 19-9 measurement, pancreatic protocol computed tomography (CT), magnetic resonance imaging (MRI), and positron emission tomography (PET)-CT. Patients were staged according to the eighth edition of the American Joint Committee on Cancer staging system.

### 2.2. Treatment

During the study period, we treated patients according to our institution’s treatment protocol for pancreatic cancer, based on the National Comprehensive Cancer Network (NCCN) guidelines [4]. Patients with resectable disease in terms of the disease extent and medical aspects underwent surgical resection. Neoadjuvant chemotherapy with or without chemoradiotherapy (CRT) followed by surgery was considered for borderline resectable tumours. In cases of locally advanced unresectable pancreatic cancer or controlled minimally metastatic disease, chemotherapy was administered for 4–6 months. X-ray therapy or PBT was also administered if there was no disease progression or only LP. Additionally, X-ray therapy or PBT was considered when surgery was unsuitable because of the patient’s medical condition, refusal to undergo surgery, or local recurrence after surgery. SBPT was preferentially considered in cases without bowel invasion at the time of diagnosis.

Before CT scan, patients underwent at least once respiratory training to breath as regularly and as shallowly as possible. The planning CT was taken on a Discovery CT590 (General Electric Healthcare, Chicago, Illinois, USA). All patients underwent four-dimensional CT (4D-CT) simulation in the supine position, with a slice thickness of 2.5 mm. After obtaining non-enhanced 4D-CT images for SBPT planning, an intravenous contrast agent was injected to improve the accuracy of delineation of the target and normal organs For planning, maximum intensity projection images were generated from 4D-CT scans and used for target delineation.

The gross tumour volume (GTV) included all identified gross lesions and metastatic lymph nodes based on diagnostic CT, MRI, PET-CT, and simulation CT findings. The clinical target volume (CTV) was defined as the GTV [19]. The internal target volume (ITV) was delineated based on all respiratory phases using 4D-CT in all patients, except two patients. These two patients were treated with a gating plan, and the ITV was delineated based on 30–70% respiratory phases. The simultaneous integrated boost (SIB) technique was used in two-thirds of the PBT plans. The low-risk planning target volume (PTV) was delineated by adding a 5-mm margin to the ITV to account for setup uncertainties. High-risk PTV was delineated by removing the planning organ-at-risk volume from the ITV by adding a 3-mm margin. In cases treated with the SIB technique, the prescribed dose to the low-risk PTV was 50–65% of the high-risk PTV according to the surrounding normal organ.

RayStation (RaySearch Laboratories, Stockholm, Sweden) was used for treatment planning. The proton plans for all patients treated with the free-breathing technique were calculated on PTVs defined from MIP images. The dose distribution and dose-volume histograms (DVHs) for calculated plan on the MIP were verified with the max- and min-inhalation phases of the 4DCT datasets. Additionally, all plans were robustly optimized to PTV using 5 mm setup uncertainty and 3.5% range uncertainty. PBT was delivered using a proton therapy system (Sumitomo, Tokyo, Japan) at Samsung Medical Center. In all cases, proton therapy was delivered using the pencil beam line-scanning method, a passive scanning technique. Line scanning proceeded in the lateral direction, and after the line reached the lateral edge, the next line was scanned in the opposite direction. Three proton beams were irradiated in the posterior–anterior, right posterior oblique, and left posterior oblique directions to reduce the effects of the range and setup uncertainty. The dose distribution in the target was optimised by adjusting line and layer spacings (Figure 1). A spot size of 70% was used for all patients because of the high-quality plan [20]. Daily cone-beam CT and/or orthogonal kilovoltage X-ray images provided by VeriSuite (MedCom, Darmstadt, Germany) were obtained and matched with simulation images mainly based on the vertebral bodies before every treatment session.

### 2.3. Endpoints and Statistical Analysis

Patients underwent regular follow-up examinations including physical examination, haematologic studies, and CT. The endpoints included OS, progression-free survival (PFS), and LC. All endpoints were calculated from the start date of SBPT to the date of the last follow-up or event. Additionally, OS and PFS were calculated from the start date of the treatment course to the last follow-up or event, respectively. Tumour response to RT was evaluated using the Revised Response Evaluation Criteria in Solid Tumours guidelines (version 1.1). The best response was recorded from treatment initiation to disease progression or recurrence. Failure patterns after SBPT were classified as locoregional, peritoneal seeding, or distant haematogenous metastases. Treatment-related toxicity was graded according to the National Cancer Institute Common Terminology Criteria for Adverse Events, version 5.0. Survival outcomes were analysed using the Kaplan–Meier method and compared using the log-rank test for univariate analysis. The Cox proportional hazard regression model was used for univariate and multivariate analyses. Statistical significance was set at *p* < 0.05. Multivariable analysis was performed on variables with a probability value of <0.2 or on those that were considered relevant. Statistical analyses were performed using SPSS Statistics version 27.0 (IBM Corp, Armonk, NY, USA).

## 3. Results

### 3.1. Patient Characteristics

The baseline characteristics of all patients are summarised in Table 1. The median patient age was 61 years (range, 49–90 years). In terms of resectability, 81.5% cases were unresectable, 10.3% were borderline resectable, and 8.2% were resectable. The head, body, and tail of the pancreas were the primary tumour locations in 23 (46.9%), 22 (44.9%), and four (8.2%) patients, respectively.

Eight (16.3%) patients underwent surgery before or after SBPT; two patients underwent surgery as they responded to induction chemotherapy and SBPT. The other patients underwent surgery during the previous treatment course.

Approximately 90% patients received induction chemotherapy prior to SBPT. Most patients (*n* = 31, 70.5%) received either FOLFIRINOX (5-fluorouracil, leucovorin, irinotecan, and oxaliplatin) or modified FOLFIRINOX therapy. The median chemotherapy to SBPT interval for these patients was 6.7 months (range, 2.1–23.4 months). Partial response (PR), stable disease (SD), and PD were observed in 32.7%, 51.0%, and 6.1% patients, respectively.

The dose fractionation scheme used was 60 GyRBE in five fractions (*n* = 42, 85.7%) or 50 GyRBE in five fractions (*n* = 7, 14.3%). The median PTV was 79.0 cm^3^ (range, 20.5–291.8 cm^3^).

### 3.2. Survival Outcomes and Prognostic Factors

The median follow-up after SBPT was 16.3 months (range, 1.8–45.0 months). The 1-year OS, PFS, and LC rates were 90.7%, 54.3% and 88.6%, respectively. The 2-year OS, PFS, and LC rates were 67.6%, 38.0% and 73.0%, respectively. Additionally, we calculated OS and PFS from the start of chemotherapy or the start of SBPT if chemotherapy was not administered. Using this calculation, the 1-year OS and PFS rates were 95.6% and 76.5%, respectively. The 2-year OS and PFS rates were 78.9% and 42.3%, respectively (Figure 2).

Age (hazard ratio, 1.10; 95% confidence interval, 1.01–1.20; *p* = 0.024) was an independent prognostic factor for PFS (Table 2). The univariate and multivariate Cox proportional hazard models for OS and LC are presented in Table 3 and Table 4, respectively.

### 3.3. Treatment Response and Patterns of Failure

Regarding anti-tumour activity, the best response after SBPT was complete response (CR) in four (8.2%) patients, PR in 13 (26.5%) patients, SD in 31 (63.3%) patients, and PD in one (2.0%) patient. During follow-up, 26 (53.1%) patients developed PD. Figure 3 shows the failure patterns after SBPT. Locoregional failure was observed in 12 (24.5%) patients. Peritoneal seeding and distant haematogenous metastases occurred in 13 (26.5%) and 14 (28.6%) patients, respectively.

### 3.4. Treatment-Related Gastroduodenal Toxicity

All patients tolerated SBPT well, but 16 experienced treatment-related GD toxicities during follow-up. Grade ≥ 3 GD toxicities occurred in three (6.1%) patients. One patient had GD perforation due to a severe GD ulcer (Appendix A). Among the 13 patients who experienced grade 1 or 2 GD toxicities, 10 improved spontaneously or with medication and three patients showed a similar degree of symptoms during follow-up. One patient with grade 3 toxicity underwent endoscopic haemostasis. Two patients with grade 4 toxicity underwent emergency surgery. One patient underwent primary repair of a gastric perforation and another underwent total gastrectomy. Postoperatively, no significant SBPT-related complications occurred during follow-up (Appendix A).

The relationship between the GD toxicities and DVH values is presented in Table 5. The dosimetric values were converted into EQD_2_ values. Overall, the dosimetric values (Dmax, D5cc, and D10cc in EQD_2_) were not significantly different between the grade ≥ 2 and lower toxicity groups.

## 4. Discussion

The role of RT in pancreatic cancer is controversial because of conflicting results, particularly in LAPC [6,21,22]. Therefore, to clarify the role of CRT in LAPC, the phase III randomised LAP07 trial compared CRT after induction chemotherapy with chemotherapy alone. CRT did not lead to increased OS, but it significantly prolonged the treatment-free period and time to LP [9,22]. Consequently, RT has been considered an acceptable treatment option for LAPC, especially in patients with controlled disease after systemic chemotherapy and/or who cannot tolerate further systemic chemotherapy according to the current American Society of Clinical Oncology (ASCO) and NCCN guideline [4,23].

However, the LC rate with conventional fractionated CRT is low (50–60%). This shifts the question from whether RT is beneficial to how to make RT more effective in pancreatic cancer. Therefore, advanced techniques, such as intensity-modulated RT or SBRT, that escalate radiation doses to primary tumours have been attracting attention [21,22].

SBRT delivers a more ablative radiation dose to the tumour volume while reducing the dose to the surrounding normal at-risk organs. Furthermore, a short treatment duration can improve quality of life and make it easier to combine with other modalities [6]. Early SBRT studies used single-fraction SBRT, reporting high 1-year LC and OS rates of 84–100% and 21–50%, respectively. However, the incidence of grade ≥ 2 late GD toxicities was also high (20–44%) [6,24,25,26,27,28]. Subsequently, multi-fractionated SBRT was used to reduce toxicity. In previous studies, it reduced late GD toxicities with comparable LC rates compared with single-fraction SBRT [6,24,29]. Mahadevan et al. analysed the treatment outcomes of SBRT with chemotherapy in 39 patients with LAPC who received SBRT of 24–30 Gy in three fractions. The median OS and PFS were 20 and 15 months, respectively. The 1-year freedom from LP (FFLP) rate was 85.0%. Grade ≥ 3 late GD toxicity occurred in only 9% patients [30]. In another recent study, Jung et al. analysed the treatment outcomes of SBRT with chemotherapy in 95 patients with LAPC who received SBRT of 24–36 Gy in four fractions. The median OS and PFS were 16.7 and 10.2 months, respectively. The 1-year OS and PFS rates were 67.4% and 42.9%, respectively. The 1-year FFLP rate was 80.1%. Grade ≥ 3 late GD toxicity occurred in only 3% cases [24]. Moreover, several other studies on fractionated SBRT have reported favourable 1-year and 2-year OS rates of 35–85% and 0–50% and a 1-year FFLP rate of 40–100%, with a minimal toxicity rate of 0–23% [29]. Based on these advantages, SBRT can be considered an alternative to conventional fractionated CRT in experienced centres.

PBT offers further clinical advantages over SBRT and conventional RT with X-ray. It has a unique depth-dose distribution with a sharp dose peak (Bragg peak) at a specific depth of tissue, which enables substantial reductions in doses delivered to the normal tissues proximal and distal to the target volume. Due to this characteristic dosimetric profile, it is possible to deliver escalated dose to the tumor and minimize low to medium dose exposed volume of normal tissues [10,29,31]. In several studies comparing proton and photon plan in pancreatic cancer, the PBT plans showed significant lower doses to normal organs like liver, duodenum, small bowel. Dose reduction to radiosensitive normal organs can translate to a potential decline in both acute and long-term RT related toxicities and consequently fewer interruptions to intensive and aggressive systemic therapy [12,13,14,15,16,17,18]. In addition, PBT can reduce radiation induced lymphopenia, known as an unfavourable prognostic factor, by decreasing low dose irradiated volume of normal organs like spine and vessels [32,33].

Terashima et al. analysed 50 patients with LAPC treated with PBT (50–70.2 GyRBE in 25–26 fractions) with concurrent gemcitabine and showed favourable outcomes in terms of the 1-year OS, recurrence-free survival (RFS), and LP-free survival rates (76.8%, 64.3% and 81.7%, respectively) and grade ≥ 3 late toxicity rate (10%) [34]. Kim et al. analysed 40 patients with localised pancreatic cancer treated with SBPT (30–45 GyRBE in 10 fractions) with or without induction chemotherapy and showed favourable outcomes in terms of 1-year OS, RFS, and FFLP rates (75.7%, 33.2% and 64.8%, respectively) and grade ≥ 3 late toxicity rate (0%) [10].

In our study, we aimed to report the survival outcomes and GD toxicities associated with SBPT in pancreatic cancer. We analysed 49 patients with pancreatic cancer treated with SBPT and prescribed 50–60 GyRBE in five fractions according to the proximity of the normal organs. The 1-year OS, PFS, and LC rates were 90.7%, 54.3% and 88.6%, respectively, while the 2-year OS, PFS, and LC rates were 67.6%, 38.0% and 73.0%, respectively. The 1- and 2-year OS and PFS rates, which were calculated from the date of treatment initiation, were 95.6% and 78.9% and 76.5% and 42.3%, respectively. These results were favourable compared to those of the LAP07 trial and those of other SBRT or proton studies [9,29]. In our study, a longer chemotherapy to RT interval (≥4 months vs. <4 months) was an independent favourable prognostic factor for OS. This result is consistent with that of a previous study that emphasises the importance of induction chemotherapy [35]. This may be because longer induction chemotherapy regimens provide sufficient time for sterilisation of micrometastases or better selection of an effective patient population for RT for local disease.

The incidence of grade ≥ 3 GD toxicities was 6.1% (*n* = 3), which is similar to that reported in previous SBRT studies (0–15.6%). Considering that the prescribed doses to the target volume were higher than those in most previous SBRT studies that administered 20–50 Gy in three to seven fractions, SBPT had significant dosimetric advantages, with superior survival outcomes [29,30,36,37,38,39,40]. Moreover, some patients experienced severe GD toxicities, but they recovered with proper management without lasting complications.

This study has some limitations. First, this was a retrospective study with a small number of patients and a short follow-up period. Additional prospective studies with larger sample sizes and longer follow-up periods are required. Second, oesophagogastroduodenoscopy was only performed when the patient was symptomatic. Therefore, it is likely that the toxicity-reducing effect of SBPT, along with its favourable treatment outcomes, has been overestimated. Third, significant heterogeneity was observed in the patient population in terms of disease status, combined treatment regimens, RT timing, and other factors. Finally, we only included patients treated with SBPT; hence, we cannot directly compare the outcomes with those of patients who underwent treatment using other modalities such as conventional CRT or SBRT. Therefore, we cannot conclude that SBPT is superior to other RT techniques such as conventional CRT in terms of its toxicity profile. In this regard, another study comparing proton- and photon-based therapies may be helpful.

## 5. Conclusions

The present data show that SBPT can be a very effective and safe treatment option for LC in a well-selected patient population. Future studies should focus on developing an optimal scheme for chemotherapy and SBPT in terms of timing, regimen, dose, field, and other factors. Careful patient selection considering the risks and benefits should continue when administering SBPT in patients with pancreatic cancer, especially those with tumours with GD invasion.

## Figures and Tables

**Figure 1 cancers-14-04556-f001:**
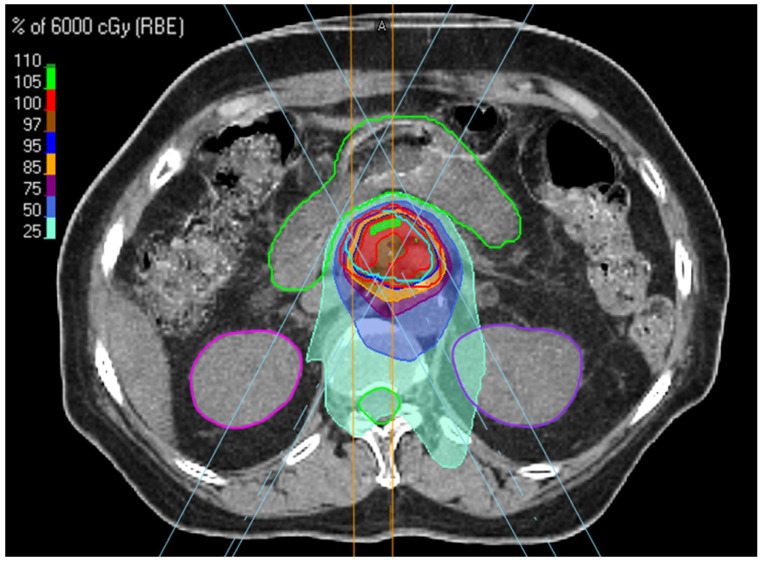
Beam arrangement and dose distribution in SBPT planning.

**Figure 2 cancers-14-04556-f002:**
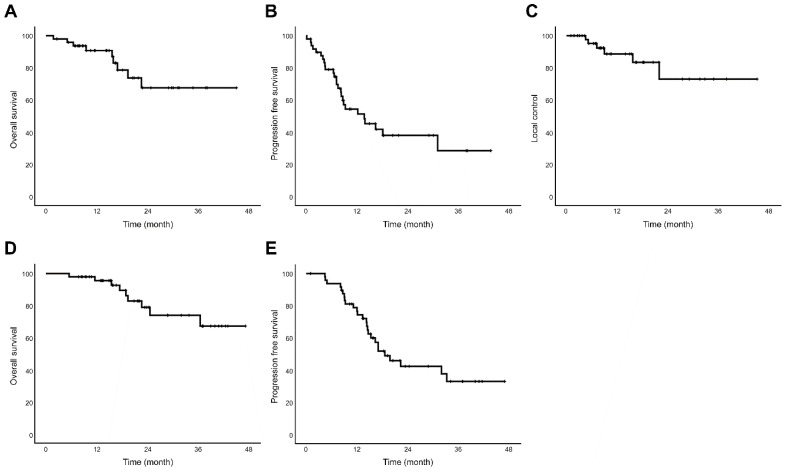
Kaplan–Meier survival curves of all enrolled patients treated with stereotactic body proton beam therapy (*n* = 49) (**A**) Overall survival (**B**) Progression-free survival (**C**) Local control rate (**D**) Overall survival from start date of treatment course (**E**) Progression-free survival from start date of treatment course.

**Figure 3 cancers-14-04556-f003:**
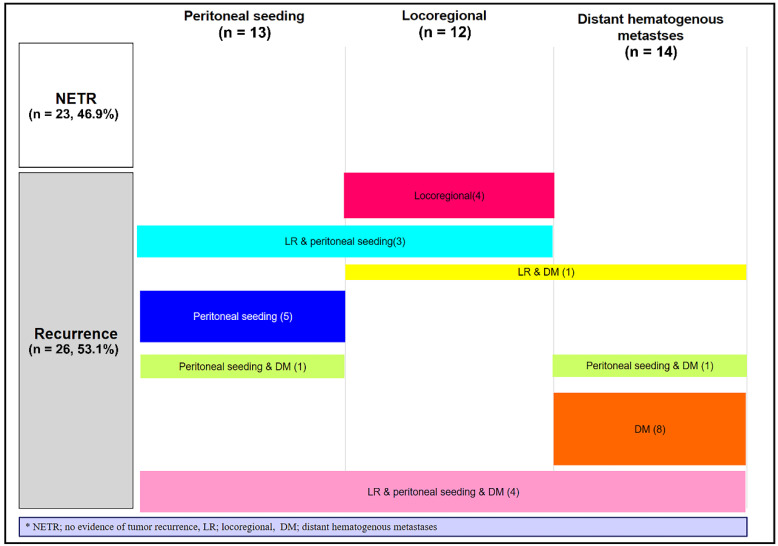
Failure patterns after stereotactic body proton beam therapy (*n* = 49).

**Table 1 cancers-14-04556-t001:** Characteristics of patients (*n* = 49).

Characteristics	N (%)
Median age (years, range)	61 (46–90)
Sex	
Male	31 (63.3)
Female	18 (36.7)
Tumor stage	
T1	0 (0)
T2	2 (4.1)
T3	9 (18.4)
T4	38 (77.6)
Lymph node metastasis	
N0	22 (44.9)
N1	23 (46.9)
N2	4 (8.2)
Distant metastasis	
No	43 (87.8)
Yes	6 (12.2)
Primary site	
Head	23 (46.9)
Body	22 (44.9)
Tail	4 (8.2)
Resectability	
Unresectable	40 (81.5)
Borderline resectable	5 (10.3)
Resectable	4 (8.2)
Pre-SBPT CA19-9 (U/mL, median, range)	65.40 (3.33–2536.75)
Pre-SBPT ALC (10^3^/μL, median, range)	1.80 (0.81–3.31)
Surgery	
No	41 (83.7)
Yes	8 (16.3)
Before SBPT	6 (12.2)
After SBPT	2 (4.1)
RT aim	
Definitive (no other previous treatment)	3 (6.1)
Consolidation (after induction chemotherapy)	38 (77.6)
Salvage	8 (16.3)
Progression after induction chemotherapy	3 (6.1)
After induction chemotherapy for recurrence	3 (6.1)
Immediate RT after recurrence	2 (4.1)
Induction chemotherapy (*n* = 44)	
FOLFIRINOX	31 (70.5)
gemcitabine/abraxane	13 (29.5)
Interval induction chemotherapy to SBPT (*n* = 44)	
<4 month	10 (20.4)
≥4 month	34 (79.6)
Induction chemotherapy response (*n* = 44)	
PR	16 (32.7)
SD	25 (51.0)
PD	3 (6.1)
Chemotherapy after SBPT	
No	9 (18.4)
Yes	40 (81.6)
FOLFIRINOX	23 (46.9)
gemcitabine/abraxane	11 (22.4)
Others *	6 (12.2)
RT dose schedule	
50 GyRBE in 5 fractions	7 (14.3)
60 GyRBE in 5 fractions	42 (85.7)
PTV (cc, median, range)	79.00 (20.50–291.80)

SBPT: Stereotactic body proton beam therapy; ALC: absolute lymphocyte count; CA 19-9: Carbohydrate antigen 19-9; RT: radiotherapy; CT: chemotherapy; PR: partial response; SD: stable disease; PD: progressive disease; GyRBE: gray relative biological effectiveness; PTV: planning target volume. Stage according to American joint cancer committee 8th edition. Tumor response to radiotherapy was evaluated using the Revised Response Evaluation Criteria in Solid Tumors guidelines (version 1.1). * gemcitabine/erlotinib (*n* = 3), gemcitabine (*n* = 1), TS-1 (tegafur, gimeracil, and oteracil potassium) (*n* = 2), capecitabine/oxaliplatin (*n* = 1), pembrolizumab (*n* = 1).

**Table 2 cancers-14-04556-t002:** Univariable and multivariable Cox proportional hazard model for progression-free survival (*n* = 49).

	No. (%)	Univariable		Multivariable	
		HR (95% CI)	*p*	HR (95% CI)	*p*
Age	61 (46–90) *	1.10 (1.01–1.20)	0.025	1.05 (1.01–1.10)	0.024
Sex					
Male	31 (63.3)	Reference	0.523		
Female	18 (36.7)	0.59 (0.12–2.94)			
T stage					
T1–3	11 (22.4)	Reference	0.655		
T4	38 (77.6)	1.63 (0.19–14.04)			
Lymph node metastases					
No	22 (44.9)	Reference	0.804		
Yes	27 (55.1)	1.22 (0.24–6.13)			
Distant metastases					
No	43 (87.8)	Reference	0.632		
Yes	6 (12.2)	0.04 (0.0–22.061.28)			
Primary site					
Head	23 (46.9)	Reference	0.546		
Body/Tail	26 (53.1)	0.61 (0.12–3.02)			
Induction CT					
No	5 (10.2)	Reference	0.375		
Yes	44 (89.8)	0.04 (0.04–3.32)			
Induction CT regimen					
FOLFIRINOX	31 (63.3)	Reference	0.838		
gemcitabine/abraxane	13 (26.5)	0.00 (0.00–4.199 E)	0.972		
No	5 (10.2)	1.94 (0.21–17.50)	0.553		
CT to RT interval					
<4 month	10 (20.4)	Reference	0.793		
≥4 month	34 (79.6)	1.34 (0.14–12.30)			
Induction CT response					
PR	16 (32.7)	Reference	0.661		
SD	25 (51.0)	161,339 (0–1.438 E)	0.960		
PD	3 (6.1)	446,977 (0–3.999 E)	0.957		
After CT					
No	9 (18.4)	Reference	0.172	Reference 3.38 (1.00–11.41)	0.050
Yes	40 (81.6)	0.28 (0.04–1.72)			
After CT regimen					
FOLFIRINOX	23 (46.9)	Reference	0.089	Reference 2.19 (0.91–5.26)	0.079
Others/No	26 (53.1)	1.99 (0.90–4.41)			
Pre- SBPT CA19-9					
<60 U/mL	24 (49.0)	Reference	0.112	Reference 2.06 (0.86–4.87)	0.101
≥60 U/mL	25 (51.0)	5.71 (0.66–48.89)			
Pre-SBPT ALC (10^3^/μL)	1.80 (0.81–3.31) *	1.84 (0.37–9.13)	0.455		

HR: hazard ratio; CI: confidence interval; CT: chemotherapy; RT: radiation therapy; PR: partial response; SD: stable disease; PD: progressive disease; SBPT: Stereotactic body proton beam therapy; CA 19-9: Carbohydrate antigen 19-9; ALC: absolute lymphocyte count. Stage according to American joint cancer committee 8th edition. Tumor response to radiotherapy was evaluated using the Revised Response Evaluation Criteria in Solid Tumors guidelines (version 1.1). * The values are presented as median (range).

**Table 3 cancers-14-04556-t003:** Univariable and multivariable Cox proportional hazard model for overall survival (*n* = 49).

	No. (%)	Univariable		Multivariable	
		HR (95% CI)	*p*	HR (95% CI)	*p*
Age	61 (46–90) *	0.96 (0.88–1.05)	0.435		
Sex					
Male	31 (63.3)	Reference	0.219		
Female	18 (36.7)	38.79 (0.11–13,265.15)			
T stage					
T1–3	11 (22.4)	Reference	0.371		
T4	38 (77.6)	2.58 (0.32–20.67)			
Lymph node metastases					
No	22 (44.9)	Reference	0.488		
Yes	27 (55.1)	1.60 (0.42–6.06)			
Distant metastases					
No	43 (87.8)	Reference	0.770		
Yes	6 (12.2)	1.36 (0.16–11.24)			
Primary site					
Head	23 (46.9)	Reference	0.049	Reference	0.074
Body/Tail	26 (53.1)	0.24 (0.05–0.99)		0.27 (0.06–1.14)	
Induction CT					
No	5 (10.2)	Reference	0.592		
Yes	44 (89.8)	23.11 (0.00–2,212,943.9)			
Induction CT regimen					
FOLFIRINOX	31 (63.3)	Reference	0.989		
gemcitabine/abraxane	13 (26.5)	1.11 (0.27–4.47)	0.881		
No	5 (10.2)	0.00 (0.00)	0.988		
CT to RT interval					
<4 month	10 (20.4)	Reference	0.173	Reference	0.296
≥4 month	34 (79.6)	0.40 (0.11–1.49)		0.49 (0.12–1.86)	
Induction CT response					
PR	16 (32.7)	Reference	0.962		
SD	25 (51.0)	1.21 (0.30–4.86)	0.782		
PD	3 (6.1)	0.00 (0.0)	0.991		
After CT					
No	9 (18.4)	Reference	0.748		
Yes	40 (81.6)	1.41 (0.17–11.42)			
After CT regimen					
FOLFIRINOX	23 (46.9)	Reference	0.331		
Others/No	26 (53.1)	1.99 (−0.69–2.07)			
Pre-SBPT CA19-9					
<60 U/mL	24 (49.0)	Reference	0.436		
≥60 U/mL	25 (51.0)	1.73 (0.43–6.94)			
Pre-SBPT ALC (10^3^/μL)	1.80 (0.81–3.31) *	0.49 (0.15–1.58)	0.235		

Abbreviations are as presented in the above table. * The values are presented as median (range).

**Table 4 cancers-14-04556-t004:** Univariate and multivariate Cox proportional hazard models for local control (*n* = 49).

	No. (%)	Univariable		Multivariable	
		HR (95% CI)	*p*	HR (95% CI)	*p*
Age	61 (46–90) *	1.05 (1.01–1.10)	0.023	1.05 (1.01–1.10)	0.023
Sex					
Male	31 (63.3)	Reference	0.919		
Female	18 (36.7)	1.04 (0.46–2.34)			
T stage					
T1–3	11 (22.4)	Reference	0.776		
T4	38 (77.6)	0.88 (0.36–2.11)			
Lymph node metastases					
No	22 (44.9)	Reference	0.331		
Yes	27 (55.1)	1.47 (0.67–3.23)			
Distant metastases					
No	43 (87.8)	Reference	0.637		
Yes	6 (12.2)	1.35 (0.39–4.61)			
Primary site					
Head	23 (46.9)	Reference	0.088		
Body/Tail	26 (53.1)	0.51 (0.23–1.10)			
Induction CT					
No	5 (10.2)	Reference	0.590		
Yes	44 (89.8)	0.71 (0.21–2.41)			
Induction CT regimen					
FOLFIRINOX	31 (63.3)	Reference	0.817		
gemcitabine/abraxane	13 (26.5)	1.16 (0.48–2.84)	0.732		
No	5 (10.2)	1.45 (0.42–5.03)	0.551		
CT to RT interval					
<4 month	10 (20.4)	Reference	0.379		
≥4 month	34 (79.6)	1634 (0.54–4.83)			
Induction CT response					
PR	16 (32.7)	Reference	0.357		
SD	25 (51.0)	1.99 (0.77–5.15)	0.155		
PD	3 (6.1)	1.87 (0.37–9.34)	0.443		
After CT					
No	9 (18.4)	Reference	0.726		
Yes	40 (81.6)	1.21 (0.41–3.51)			
After CT regimen					
FOLFIRINOX	23 (46.9)	Reference	0.086		
Others/No	26 (53.1)	6.59 (0.77–56.60)			
Pre-SBPT CA19-9					
<60 U/mL	24 (49.0)	Reference	0.067		
≥60 U/mL	25 (51.0)	2.13 (0.94–4.79)			
Pre-SBPT ALC (10^3^/μL)	1.80 (0.81–3.31) *	0.81 (0.41–1.60)	0.556		

Abbreviations are as presented in the above table. * The values are presented as median (range).

**Table 5 cancers-14-04556-t005:** Relationship between dosimetric parameters and Grade 2 or higher gastroduodenal toxicities in stereotactic body proton beam therapy (*n* = 49).

Parameter	Cut off Value of EQD_2_	HR	95% CI	*p*
Dmax	≥57.000 GyRBE	2.11	0.41–10.98	0.373
D5cc	≥10.028 GyRBE	2.44	0.58–10.37	0.225
D10cc	≥5.990 GyRBE	2.16	0.50–9.22	0.296

SBPT: Stereotactic body proton beam therapy; EQD_2_: the equivalent dose in 2 Gy fractions; HR: hazard ratio; CI: confidence interval; GyRBE: gray relative biological effectiveness. Event included Grae2 or higher gastroduodenal toxicities. Dmax means maximal dose delivered to the organ. D‘n’cc means dose delivered to ‘n’ cc of organ volume. EQD_2_ was calculated with alpha beta ratio of 3.

## Data Availability

Data availability is limited due to institutional data protection law and confidentiality of patient data.

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
