# Peer review of "The Feasibility of Stereotactic Body Proton Beam Therapy for Pancreatic Cancer"

_cancers, 2022, doi:10.3390/cancers14194556_

Round 1

Reviewer 1 Report

I am grateful for the opportunity to review manuscript ID: cancers-1887510, entitled “The efficacy and safety of stereotactic body proton beam therapy in pancreatic cancer”.

The authors report outcome data from a retrospective analysis of 49 patients treated with SBPT for pancreatic cancer between 2017 and 2020. This is an interesting area of investigation, as radiotherapy for pancreatic cancer is notoriously challenging with limited benefit, but the potential for improved outcomes with PBT is attractive. Taking this a step further to investigate SBPT is worthwhile.

The overall manuscript is clearly written, and patient related data and study findings are well presented. The major limitations with this study are mentioned by the authors in the discussion. These limitations do hinder the ability to conclude that SBPT is more efficacious than other RT/PBT modalities, although I do see the value in publishing this initial work to highlight at least the feasibility. I think that this work is of interest to the readership, however I have some comments:

·        Treatment planning was based on 4DCT simulation. The plans were calculated on the maximum intensity projection (MIP) datasets, which will over estimate the required proton ranges. An alternative approach is to contour on the MIP and calculate on the AVG dataset. The authors should at least verify the plan as calculated on the MIP on, say, the max- and min-inhalation phases of the 4DCT datasets. A sub-set of cases could be analyzed in this way for the purposes of this study.

·        The authors describe the GTV, and then state that the CTV is the same as the GTV. Could the authors clarify that indeed no expansion of the GTV to a CTV was included, and explain why?

·        The PTV included a 5mm expansion of the ITV for setup uncertainty. How were beam direction range uncertainties delt with? These margins are independent of setup uncertainty margins.

·        Robust optimization using 5mm/3% for setup and range would have appropriate, although I suspect robust optimization was not available at the time of the initial planning. Did the authors perform robust analysis for setup and range of the nominal treatment plan? If not, why not?

·        Besides accounting for motion with 4DCT-based planning, were any other motion mitigation techniques used during simulation and treatment delivery?

·        Beam angles used were PA, RPO, and LPO. A figure showing a typical setup and dose distribution would be a useful addition.

·        Given the limitations of this study, and their impact on conclusions that may be drawn, I suggest revising the title of this manuscript. Perhaps: “The feasibility of stereotactic body proton beam therapy for pancreatic cancer”.

·        The Simple Summary has some grammatical errors throughout. (The manuscript overall is well written, however).

·        Some acronyms throughout the manuscript need defining, and the CT scanner model/manufacturer should be included.

Author Response

Reviewer #1

I am grateful for the opportunity to review manuscript ID: cancers-1887510, entitled “The efficacy and safety of stereotactic body proton beam therapy in pancreatic cancer”.

The authors report outcome data from a retrospective analysis of 49 patients treated with SBPT for pancreatic cancer between 2017 and 2020. This is an interesting area of investigation, as radiotherapy for pancreatic cancer is notoriously challenging with limited benefit, but the potential for improved outcomes with PBT is attractive. Taking this a step further to investigate SBPT is worthwhile.

The overall manuscript is clearly written, and patient related data and study findings are well presented. The major limitations with this study are mentioned by the authors in the discussion. These limitations do hinder the ability to conclude that SBPT is more efficacious than other RT/PBT modalities, although I do see the value in publishing this initial work to highlight at least the feasibility. I think that this work is of interest to the readership, however I have some comments:

Treatment planning was based on 4DCT simulation. The plans were calculated on the maximum intensity projection (MIP) datasets, which will over estimate the required proton ranges. An alternative approach is to contour on the MIP and calculate on the AVG dataset. The authors should at least verify the plan as calculated on the MIP on, say, the max- and min-inhalation phases of the 4DCT datasets. A sub-set of cases could be analyzed in this way for the purposes of this study.

→ First of all, we sincerely appreciate your kind and detailed review. We tried to make corrections to reflect your insightful feedback.

We verifies the dose distribution on the max- and min-inhalation phases of 4DCT datasets with the plan of all patients treated with the free breathing technique. Therefore, we added sentences in L146 as recommended.

“The proton plans for all patients treated with the free-breathing technique were calculated on PTVs defined from MIP images. The dose distribution and dose-volume histograms (DVHs) for calculated plan on the MIP were verified with the max- and min-inhalation phases of the 4DCT datasets.”

 The authors describe the GTV, and then state that the CTV is the same as the GTV. Could the authors clarify that indeed no expansion of the GTV to a CTV was included, and explain why?

→ We gave no additional GTV to CTV margin as ESTRO ACROP guidelines [1]. We think that the main role of SBPT is to control gross tumor and intensive systemic therapy to control microscopic disease in pancreatic cancer. And growing evidence from several tumor models such as non-small cell lung cancer suggests that small involved field radiation may be reasonable without compromising locoregional control and overall survival.[2] We added ESTRO ACROP guideline to the reference lists (L135).

The PTV included a 5mm expansion of the ITV for setup uncertainty. How were beam direction range uncertainties dealt with? These margins are independent of setup uncertainty margins.

Robust optimization using 5mm/3% for setup and range would have appropriate, although I suspect robust optimization was not available at the time of the initial planning. Did the authors perform robust analysis for setup and range of the nominal treatment plan? If not, why not?

→ We dealt with the plan optimization applying 3.5% range uncertainties for beam direction. Therefore, we added a sentence in L149 as recommended.

“Additionally, all plan were robustly optimized to PTV using 5 mm setup uncertainty and 3.5% range uncertainty.”

Besides accounting for motion with 4DCT-based planning, were any other motion mitigation techniques used during simulation and treatment delivery?

→ We educated patients for regular and shallow breathing before CT scanning. We added the sentences in L125

Beam angles used were PA, RPO, and LPO. A figure showing a typical setup and dose distribution would be a useful addition.

→Thank you for your feedback. We think it would be better to add a figure for planning, too.

We added ‘Figure 1’ for beam angles as you recommended in L164, and renamed original ‘Figure 1’ to ‘Figure 2’, ‘Figure 2 to Figure 3’.

Given the limitations of this study, and their impact on conclusions that may be drawn, I suggest revising the title of this manuscript. Perhaps: “The feasibility of stereotactic body proton beam therapy for pancreatic cancer”.

→ Thank you for your thoughtful comments. We changed the title as your feedback

The Simple Summary has some grammatical errors throughout. (The manuscript overall is well written, however).

→ Thank you for finding my grammatical error. We corrected the error in simple summary as below

“Despite advances in treatment, the treatment outcome of pancreatic cancer still remains poor. Local progression can be a significant cause of several morbidities in pancreatic cancer, and dose escalation is needed. Stereotactic body proton beam therapy (SBPT)  can give higher dose while minimizing dose at organ at risk with Bragg peak. Purpose of the present study was to investigate the feasibility of SBPT in pancreatic cancer. SBPT, administered in five fractions of a total 50–60 GyRBE, was performed mostly after induction chemotherapy. Grade 3 or higher gastroduodenal toxicities occurred in 6.1% of cases. The 2-year overall survival and local control rates were 67.6% and 73.0%. SBPT showed favorable survival outcomes and treatment-related toxicities. It could be a promising alternative to radical surgery.”

Some acronyms throughout the manuscript need defining, and the CT scanner model/manufacturer should be included.

→ Thank you for your thoughtful review. We added defining of some acronyms in L101, L193, L208, and L230. We inserted the information about CT scanner model and manufacturer used for radiation therapy simulation in L125.

Reviewer 2 Report

The tables need to put the types of chemotherapy "under" the total chemotherapy number. It is obvious upon close reading that the types all add up to the total percentage of those getting chemotherapy, but it would be cleaner to make it more clear via the tables' formatting.   Images of the plan might be of interest to show how you stayed off the duodenum, etc. 

Author Response

The tables need to put the types of chemotherapy "under" the total chemotherapy number. It is obvious upon close reading that the types all add up to the total percentage of those getting chemotherapy, but it would be cleaner to make it more clear via the tables' formatting. 

→ First of all, thank you very much for taking the time to review our article. We have tried to revise the article in response to your sincere feedback.

We agree with your opinion that it would be clearer to put the types of chemotherapy under the total chemotherapy number. We corrected the corresponding contents in table 1.

Images of the plan might be of interest to show how you stayed off the duodenum, etc.

→ Thank you for the good point. We added ‘Figure 1’ for beam arrangement and dose distribution and renamed original ‘Figure 1’ to ‘Figure 2’and ‘Figure 2 to Figure 3’.

Round 2

Reviewer 1 Report

With this revised submission, the authors have adequately addressed reviewer comments.

Author Response

The authors have improved the tables and satisfactorily addressed the reviewed comments.  Also this is a modestly sized retrospective cohort, the report adds considerably to the literature on proton SBRT in general, and specifically for pancreatic cancer.  As such, the authors should be commended, and this work will be suitable for publication.  First, however, the authors discuss the potential benefits of radiation therapy and of SBRT at length, but do a much more limited job of discussing the potential benefits of proton therapy.  They should discuss other work studies, including the reviews on proton therapy for pancreatic cancer (including a recent review published in this journal (examples: Cancers (Basel). 2022 Jun 4;14(11):2789. and J Gastrointest Oncol. 2016 Aug;7(4):644-64.) and also some of the key studies cited in those reviews.  It would be beneficial to add a sentence or two towards the end of the introduction on this, which will serve as a rationale for analyzing their cohort.  And then adding a bit more detail in the Discussion section.  Also, reference #3 is dated and should be updated.

  • Thanks for your insightful feedback.
  • We agree with you that description of the benefit of SBPT was lacking compared to those describing the advantages of SBRT. We added additional sentences and references in L87(introduction part) and L311(discussion part) as below
  • “Several studies have demonstrated that the radiation exposure of normal organs including liver, duodenum, and small bowel can be reduced using proton beam than photon beam in the treatment planning”
  • “It has a unique depth-dose distribution with a sharp dose peak (Bragg peak) at a specific depth of tissue, which enables substantial reductions in doses delivered to the normal tissues proximal and distal to the target volume. Due to this characteristic dosimetric profile, it is possible to deliver escalated dose to the tumor and minimize low to medium dose exposed volume of normal tissues [11,30,32]. In several studies comparing proton and photon plan in pancreatic cancer, the PBT plans showed significant lower doses to normal organs like liver, duodenum, small bowel. Dose reduction to radiosensitive normal organs can translate to a potential decline in both acute and long term RT related toxicities and consequently fewer interruptions to intensive and aggressive systemic therapy [13-19]. In addition, PBT can reduce radiation induced lymphopenia, known as an unfavourable prognostic factor, by decreasing low dose irradiated volume of normal organs like spine and vessels [33,34].”
  • Also, we changed the references #2 and #3 to the updated versions.
